# The Implicit Metropolis-Hastings Algorithm

**Kirill Neklyudov**
Samsung-HSE Laboratory
HSE,* Moscow, Russia
Samsung AI Center Moscow
`k.necludov@gmail.com`

**Evgenii Egorov**
Skoltech,† Moscow, Russia
`egorov.evgenyy@ya.ru`

**Dmitry Vetrov**
Samsung-HSE Laboratory
HSE,* Moscow, Russia
Samsung AI Center Moscow
`vetrovd@yandex.ru`

## Abstract

Recent works propose using the discriminator of a GAN to filter out unrealistic samples of the generator. We generalize these ideas by introducing the implicit Metropolis-Hastings algorithm. For any implicit probabilistic model and a target distribution represented by a set of samples, implicit Metropolis-Hastings operates by learning a discriminator to estimate the density-ratio and then generating a chain of samples. Since the approximation of density ratio introduces an error on every step of the chain, it is crucial to analyze the stationary distribution of such chain. For that purpose, we present a theoretical result stating that the discriminator loss upper bounds the total variation distance between the target distribution and the stationary distribution. Finally, we validate the proposed algorithm both for independent and Markov proposals on CIFAR-10, CelebA and ImageNet datasets.

## 1 Introduction

Learning a generative model from an *empirical* target distribution is one of the key tasks in unsupervised machine learning. Currently, Generative Adversarial Networks (GANs) (Goodfellow et al., 2014) are among the most successful approaches in building such models. Unlike conventional sampling techniques, such as Markov Chain Monte-Carlo (MCMC), they operate by learning the *implicit* probabilistic model, which allows for sampling but not for a density evaluation. Due to the availability of large amounts of empirical data, GANs find many applications in computer vision: image super-resolution (Ledig et al., 2017), image inpainting (Yu et al., 2018), and learning representations (Donahue et al., 2016).

Despite the practical success, GANs remain hard for theoretical analysis and do not provide any guarantees on the learned model. For now, most of the theoretical results assume optimality of the learned discriminator (critic) what never holds in practice (Goodfellow et al., 2014; Nowozin et al., 2016; Arjovsky et al., 2017). Moreover, there is empirical evidence that GANs do not learn to sample from a target distribution (Arora & Zhang, 2017).

Recently, the idea of a GAN postprocessing by filtering the generator was proposed in several works. Under the assumption that the learned discriminator evaluates the exact density-ratio they filter samples from a generator by rejection sampling (Azadi et al., 2018) or by the independent Metropolis-Hastings algorithm (Neklyudov et al., 2018; Turner et al., 2018). Since the assumption of the discriminator optimality never holds in practice, we still cannot be sure that the resulting distribution will be close to the target, we even cannot guarantee that we will improve the output of the generator.

In this work, we present a theoretical result that justifies the heuristic proposed by Neklyudov et al. (2018); Turner et al. (2018) and generalize the proposed algorithm to the case of any implicit

probabilistic models — both independent and Markov. To do that, we consider some, maybe not optimal, discriminator in the Metropolis-Hastings test, and approach the problem from the MCMC perspective. Under reasonable assumptions, we derive an upper bound on the total variation distance between the target distribution and the stationary distribution of the produced chain, that can be minimized w.r.t. parameters of the discriminator.

On real-world datasets (CIFAR-10, CelebA, ImageNet), we validate our approach using different deep generative models as independent proposals: DCGAN (Radford et al., 2015); Wasserstein GAN with gradient penalty (Gulrajani et al., 2017); variational auto-encoder (Kingma & Welling, 2014); BigGAN (Brock et al., 2018); MMD-GAN (Li et al., 2017). Every model is learned independently optimizing its original objective, what allows us to test the algorithm on a wide range of different proposals. For every proposal, we learn the discriminator from scratch (except BigGAN) and observe monotonous improvements of metrics throughout the learning. Further, we construct a Markov proposal by traversing the latent space of WPGAN generator via a Markov chain. Our experiments demonstrate that this proposal compares favorably against the independent proposal while using the same generator network.

We consider the provided theoretical analysis and the empirical evaluation as a result that allows to alleviate or even eliminate the bias of any generative model learned from the empirical distribution. To be more factual, we summarize our main contributions as follows.

- We propose the implicit Metropolis-Hastings algorithm, that can be seen as an adaptation of the classical Metropolis-Hastings algorithm to the case of an implicit probabilistic model and an empirical target distribution (Section 3).

- We justify the algorithm proposed by Neklyudov et al. (2018) and Turner et al. (2018). In particular, we demonstrate that learning the discriminator via the binary cross-entropy minimizes an upper bound on the distance between the target distribution and the stationary distribution of the chain (Section 3.5).

- We empirically validate the obtained theoretical result on real-world datasets (CIFAR-10, CelebA, ImageNet) (Section 4.1). We also demonstrate empirical gains by applying our algorithm for Markov proposals (Section 4.2).

## 2 Background

### 2.1 The Metropolis-Hastings algorithm

The MH algorithm allows for sampling from an analytic target distribution $p(x)$ by filtering samples from a proposal distribution $q(x \,|\, y)$ that is also given in the analytic form. It operates by sampling a chain of correlated samples that converge in distribution to the target (see Algorithm 1).

---

**Algorithm 1** The Metropolis-Hastings algorithm

**input** density of target distribution $\hat{p}(x) \propto p(x)$
**input** proposal distribution $q(x \,|\, y)$
   $y \leftarrow$ random init
   **for** $i = 0 \ldots n$ **do**
      sample proposal point $x \sim q(x \,|\, y)$
      $P = \min\{1, \frac{\hat{p}(x)q(y \,|\, x)}{\hat{p}(y)q(x \,|\, y)}\}$
      $x_i = \begin{cases} x, & \text{with probability } P \\ y, & \text{with probability } (1 - P) \end{cases}$
      $y \leftarrow x_i$
   **end for**
**output** $\{x_0, \ldots, x_n\}$

---

**Algorithm 2** Metropolis-Hastings GAN

**input** target dataset $\mathcal{D}$
**input** learned generator $q(x)$, discriminator $d(\cdot)$
   $y \sim \mathcal{D}$ initialize from the dataset
   **for** $i = 0 \ldots n$ **do**
      sample proposal point $x \sim q(x)$
      $P = \min\{1, \frac{d(x)(1-d(y))}{(1-d(x))d(y)}\}$
      $x_i = \begin{cases} x, & \text{with probability } P \\ y, & \text{with probability } (1 - P) \end{cases}$
      $y \leftarrow x_i$
   **end for**
**output** $\{x_0, \ldots, x_n\}$

---

If we take a proposal distribution that is not conditioned on the previous point, we will obtain the **independent** MH algorithm. It operates in the same way, but samples all of the proposal points independently $q(x \,|\, y) = q(x)$.

## 2.2 Metropolis-Hastings GAN

Recent works (Neklyudov et al., 2018; Turner et al., 2018) propose to treat the generator of a GAN as an independent proposal distribution $q(x)$ and perform an approximate Metropolis-Hastings test via the discriminator. Authors motivate this approximation by the fact that the optimal discriminator evaluates the true density-ratio

$$d^*(x) = \frac{p(x)}{p(x) + q(x)} = \arg\min_d \left[ -\mathbb{E}_{x \sim p(x)} \log d(x) - \mathbb{E}_{x \sim q(x)} \log(1 - d(x)) \right]. \quad (1)$$

Substituting the optimal discriminator in the acceptance test, one can obtain the Metropolis-Hastings correction of a GAN, that is described in Algorithm 2.

In contrast to the previous works, we take the non-optimality of the discriminator as given and analyze the stationary distribution of the resulting chain for both independent and Markov proposals. In Section 3, we formulate the implicit Metropolis-Hastings algorithm and derive an upper bound on the total variation distance between the target distribution and the stationary distribution of the chain. Then, in Appendix F, we justify Algorithm 2 by relating the obtained upper bound with the binary cross-entropy.

## 3 The Implicit Metropolis-Hastings Algorithm

In this section, we describe the implicit Metropolis-Hastings algorithm and present a theoretical analysis of its stationary distribution.

The Implicit Metropolis-Hastings algorithm is aimed to sample from an empirical target distribution $p(x)$, $x \in \mathbb{R}^D$, while being able to sample from an implicit proposal distribution $q(x \mid y)$. Given a discriminator $d(x, y)$, it generates a chain of samples as described in Algorithm 3.

We build our reasoning by first assuming that the chain is generated using some discriminator and then successively introducing conditions on the discriminator and upper bounding the distance between the chain and the target. Finally, we come up with an upper bound that can be minimized w.r.t. parameters of the discriminator.

---

**Algorithm 3**

The implicit Metropolis-Hastings algorithm

---

**input** target dataset $\mathcal{D}$
**input** implicit model $q(x \mid y)$
**input** learned discriminator $d(\cdot, \cdot)$
   $y \sim \mathcal{D}$ initialize from the dataset
   **for** $i = 0 \ldots n$ **do**
      sample proposal point $x \sim q(x \mid y)$
      $P = \min\{1, \frac{d(x,y)}{d(y,x)}\}$
      $x_i = \begin{cases} x, & \text{with probability } P \\ y, & \text{with probability } (1 - P) \end{cases}$
      $y \leftarrow x_i$
   **end for**
**output** $\{x_0, \ldots, x_n\}$

---

Here we consider the case of an implicit Markov proposal, but all of the derivations also hold for independent proposals.

The transition kernel of the implicit Metropolis-Hastings algorithm is

$$t(x \mid y) = q(x \mid y) \min\left\{1, \frac{d(x,y)}{d(y,x)}\right\} + \delta(x - y) \int dx' q(x' \mid y) \left(1 - \min\left\{1, \frac{d(x',y)}{d(y,x')}\right\}\right). \quad (2)$$

Firstly, we require the proposal distribution $q(x \mid y)$ and the discriminator $d(x, y)$ to be *continuous* and *positive* on $\mathbb{R}^D \times \mathbb{R}^D$. In Appendix A, we show that these requirements guarantee the following properties of the transition kernel $t$:

- the kernel $t$ defines a correct conditional distribution;
- the Markov chain defined by $t$ is *irreducible*;
- the Markov chain defined by $t$ is *aperiodic*.

To ensure the existence of the unique invariant probabilistic measure of the chain, we should assume the recurrence of the chain (Theorem 10.0.1, Meyn & Tweedie (2012)). We satisfy the assumption on the recurrence by introducing the minorization condition in the next subsection (Orey, 1971). Then the aforementioned properties imply the convergence of the Markov chain defined by the transition kernel $t(x \mid y)$ to the stationary distribution $t_\infty$ (Theorem 4, Roberts et al. (2004)) from any point $y$.

Further, we want the stationary distribution $t_\infty$ of our Markov chain to be as close as possible to the target distribution $p$. To measure the closeness of distributions, we consider a standard metric for analysis in MCMC — the *total variation distance*

$$\|t_\infty - p\|_{TV} = \frac{1}{2} \int |t_\infty(x) - p(x)| dx. \tag{3}$$

We assume the proposal $q(x\,|\,y)$ to be given, but different $d(x,y)$ may lead to different $t_\infty$. That is why we want to derive an upper bound on the distance $\|t_\infty - p\|_{TV}$ and minimize it w.r.t. parameters of the discriminator $d(x,y)$. We derive this upper bound in three steps in the following subsections.

### 3.1 Fast convergence

In practice, estimation of the stationary distribution $t_\infty$ by running a chain is impossible. Nevertheless, if we know that the chain converges fast enough, we can upper bound the distance $\|t_\infty - p\|_{TV}$ using the distance $\|t_1 - p\|_{TV}$, where $t_1$ is the one-step distribution $t_1(x) = \int t(x\,|\,y)t_0(y)dy$, and $t_0$ is some initial distribution of the chain.

To guarantee fast convergence of a chain, we propose to use the *minorization condition* (Roberts et al., 2004). For a transition kernel $t(x\,|\,y)$, it requires that exists such $\varepsilon > 0$ and a distribution $\nu$ that the following condition is satisfied

$$t(x\,|\,y) > \varepsilon\nu(x) \ \ \forall (x,y) \in \mathbb{R}^D \times \mathbb{R}^D. \tag{4}$$

When a transition kernel satisfies the minorization condition, the Markov chain converges "fast" to the stationary distribution. We formalize this statement in the following Proposition.

**Proposition 1** *Consider a transition kernel $t(x\,|\,y)$ that satisfies the minorization condition $t(x\,|\,y) > \varepsilon\nu(x)$ for some $\varepsilon > 0$, and distribution $\nu$. Then the distance between two consequent steps decreases as:*

$$\|t_{n+2} - t_{n+1}\|_{TV} \le (1 - \varepsilon)\|t_{n+1} - t_n\|_{TV}, \tag{5}$$

*where distribution $t_{k+1}(x) = \int t(x\,|\,y)t_k(y)dy$.*

This result could be considered as a corollary of Theorem 8 in Roberts et al. (2004). For consistency, we provide an independent proof of Proposition 1 in Appendix B.

To guarantee minorization condition of our transition kernel $t(x\,|\,y)$, we require the proposal $q(x\,|\,y)$ to satisfy minorization condition with some constant $\varepsilon$ and distribution $\nu$ (note that for an independent proposal, the minorization condition holds automatically with $\varepsilon = 1$). Also, we limit the range of the discriminator as $d(x,y) \in [b,1]\ \forall x,y$, where $b$ is some positive constant that can be treated as a hyperparameter of the algorithm. These requirements imply

$$t(x\,|\,y) \ge bq(x\,|\,y) > b\varepsilon\nu(x). \tag{6}$$

Using Proposition 1 and minorization condition (6) for $t$, we can upper bound the TV-distance between an initial distribution $t_0$ and the stationary distribution $t_\infty$ of implicit Metropolis-Hastings.

$$\|t_\infty - t_0\|_{TV} \le \sum_{i=0}^\infty \|t_{i+1} - t_i\|_{TV} \le \sum_{i=0}^\infty (1 - b\varepsilon)^i \|t_1 - t_0\|_{TV} = \frac{1}{b\varepsilon}\|t_1 - t_0\|_{TV} \tag{7}$$

Taking the target distribution $p(x)$ as the initial distribution $t_0(x)$ of our chain $t(x\,|\,y)$, we reduce the problem of estimation of the distance $\|t_\infty - p\|_{TV}$ to the problem of estimation of the distance $\|t_1 - p\|_{TV}$:

$$\|t_\infty - p\|_{TV} \le \frac{1}{b\varepsilon}\|t_1 - p\|_{TV} = \frac{1}{b\varepsilon} \cdot \frac{1}{2} \int dx \left| \int t(x\,|\,y)p(y)dy - p(x) \right|. \tag{8}$$

However, the estimation of this distance raises two issues. Firstly, we need to get rid of the inner integral $\int t(x\,|\,y)p(y)dy$. Secondly, we need to bypass the evaluation of densities $t(x\,|\,y)$ and $p(x)$. We address these issues in the following subsections.

## 3.2 Dealing with the integral inside of the nonlinearity

For now, assume that we have access to the densities $t(x \mid y)$ and $p(x)$. However, evaluation of the density $t_1(x)$ is an infeasible problem in most cases. To estimate $t_1(x)$, one would like to resort to the Monte-Carlo estimation:

$$t_1(x) = \int t(x \mid y)p(y)dy = \mathbb{E}_{y \sim p(y)}t(x \mid y). \tag{9}$$

However, straightforward estimation of $t_1(x)$ results in a biased estimation of $\|t_1 - p\|_{TV}$, since the expectation is inside of a nonlinear function. To overcome this problem, we upper bound this distance in the following proposition.

**Proposition 2** *For the kernel $t(x \mid y)$ of the implicit Metropolis-Hastings algorithm, the distance between initial distribution $p(x)$ and the distribution $t_1(x)$ has the following upper bound*

$$\|t_1 - p\|_{TV} \leq 2 \left\| q(y \mid x)p(x) - q(x \mid y)p(y)\frac{d(x,y)}{d(y,x)} \right\|_{TV}, \tag{10}$$

*where the TV-distance on the right side is evaluated in the joint space $(x,y) \in \mathbb{R}^D \times \mathbb{R}^D$.*

For the proof of this proposition, see Appendix C. Note that the obtained upper bound no longer requires evaluation of an integral inside of a nonlinear function. Moreover, the right side of (10) has a reasonable motivation since it is an averaged $l_1$ error of the density ratio estimation.

$$\left\| q(y \mid x)p(x) - q(x \mid y)p(y)\frac{d(x,y)}{d(y,x)} \right\|_{TV} = \frac{1}{2}\int p(y)q(x \mid y)\left| \frac{q(y \mid x)p(x)}{q(x \mid y)p(y)} - \frac{d(x,y)}{d(y,x)} \right| dxdy \tag{11}$$

In this formulation, we see that we still could achieve zero value of $\|t_1 - p\|_{TV}$ if we could take such discriminator that estimates the desired density ratio $\frac{d(x,y)}{d(y,x)} = \frac{q(y \mid x)p(x)}{q(x \mid y)p(y)}$.

## 3.3 Dealing with the evaluation of densities

For an estimation of the right side of (10), we still need densities $p(x)$ and $q(x \mid y)$. To overcome this issue, we propose to upper bound the obtained TV distance via KL-divergence. Then we show that obtained KL divergence decomposes into two terms: the first term requires evaluation of densities but does not depend on the discriminator $d(x,y)$, and the second term can be estimated only by evaluation of $d(x,y)$ on samples from $p(x)$ and $q(x \mid y)$.

To upper bound the TV-distance $\|\alpha - \beta\|_{TV}$ via KL-divergence $\mathrm{KL}(\alpha\|\beta)$ one can use well-known Pinsker's inequality:

$$2\|\alpha - \beta\|_{TV}^2 \leq \mathrm{KL}(\alpha\|\beta). \tag{12}$$

However, Pinsker's inequality assumes that both $\alpha$ and $\beta$ are distributions, while it is not always true for function $q(x \mid y)p(y)\frac{d(x,y)}{d(y,x)}$ in (10). In the following proposition, we extend Pinsker's inequality to the case when one of the functions is not normalized.

**Proposition 3** *For a distribution $\alpha(x)$ and some positive function $f(x) > 0 \ \forall x$ the following inequality holds:*

$$\|\alpha - f\|_{TV}^2 \leq \left(\frac{2C_f + 1}{6}\right)(\widehat{\mathrm{KL}}(\alpha\|f) + C_f - 1), \tag{13}$$

*where $C_f$ is the normalization constant of function $f$: $C_f = \int f(x)dx$, and $\widehat{\mathrm{KL}}(\alpha\|f)$ is the formal evaluation of the KL divergence*

$$\widehat{\mathrm{KL}}(\alpha\|f) = \int \alpha(x)\log\frac{\alpha(x)}{f(x)}dx. \tag{14}$$

The proof of the proposition is in Appendix D.

Now we use this proposition to upper bound the right side of (10):

$$\left\| q(y \mid x)p(x) - q(x \mid y)p(y)\frac{d(x,y)}{d(y,x)} \right\|^2_{TV} \leq \tag{15}$$

$$\leq \left(\frac{2C+1}{6}\right)\left(\widehat{\mathrm{KL}}\left(q(y \mid x)p(x) \Big\| q(x \mid y)p(y)\frac{d(x,y)}{d(y,x)}\right) + C - 1\right).$$

Here $C$ is the normalization constant of $q(x \mid y)p(y)\frac{d(x,y)}{d(y,x)}$. For the multiplicative term $(2C+1)/6$, we upper bound $C$ as

$$C = \int q(x \mid y)p(y)\frac{d(x,y)}{d(y,x)}dxdy \leq \int q(x \mid y)p(y)\frac{1}{b}dxdy = \frac{1}{b}, \tag{16}$$

since we limit the range of the discriminator as $d(x,y) \in [b,1]\ \ \forall x,y$.

Summing up the results (8), (10), (15), (16), we obtain the final upper bound as follows.

$$\|t_\infty - p\|^2_{TV} \leq \frac{1}{b^2\varepsilon^2}\|t_1 - p\|^2_{TV} \leq \frac{4}{b^2\varepsilon^2}\left\| q(y \mid x)p(x) - q(x \mid y)p(y)\frac{d(x,y)}{d(y,x)} \right\|^2_{TV} \leq \tag{17}$$

$$\leq \left(\frac{4+2b}{3\varepsilon^2 b^3}\right)\left(\underbrace{\mathbb{E}_{\substack{x \sim p(x) \\ y \sim q(y \mid x)}}\left[\log \frac{d(y,x)}{d(x,y)} + \frac{d(y,x)}{d(x,y)}\right] - 1}_{\text{loss for the discriminator}} + \mathrm{KL}\left(q(y \mid x)p(x)\Big\|q(x \mid y)p(y)\right)\right)$$

Minimization of the resulting upper bound w.r.t. the discriminator $d(x,y)$ is equivalent to the following optimization problem:

$$\min_d \mathbb{E}_{\substack{x \sim p(x) \\ y \sim q(y \mid x)}}\left[\log \frac{d(y,x)}{d(x,y)} + \frac{d(y,x)}{d(x,y)}\right]. \tag{18}$$

Thus, we derive the loss function that we can unbiasedly estimate and minimize w.r.t. parameters of $d(x,y)$. We analyze the optimal solution in the following subsection.

### 3.4 The optimal discriminator

By taking the derivative of objective (18), we show (see Appendix E) that the optimal discriminator $d^*$ must satisfy

$$\frac{d^*(x,y)}{d^*(y,x)} = \frac{q(y \mid x)p(x)}{q(x \mid y)p(y)}. \tag{19}$$

When the loss function (18) achieves its minimum, it becomes

$$\mathbb{E}_{\substack{x \sim p(x) \\ y \sim q(y \mid x)}}\left[\log \frac{q(x \mid y)p(y)}{q(y \mid x)p(x)} + \frac{q(x \mid y)p(y)}{q(y \mid x)p(x)}\right] = -\mathrm{KL}\left(q(y \mid x)p(x)\Big\|q(x \mid y)p(y)\right) + 1 \tag{20}$$

Substituting this equation into (17), we achieve $\|t_\infty - p\|_{TV} = 0$. However, since we limit the range of the discriminator $d(x,y) \in [b,1]$, the optimal solution could be achieved only when the density-ratio lies in the following range:

$$\forall x,y \quad \frac{q(y \mid x)p(x)}{q(x \mid y)p(y)} \in [b, b^{-1}]. \tag{21}$$

Therefore, $b$ should be chosen small enough that range $[b, b^{-1}]$ includes all the possible values of density-ratio. Such $b > 0$ exists if the support of the target distribution is *compact*. Indeed, if we have positive $p(x)$ and $q(x \mid y)$ on compact support, we can find a minimum of the density-ratio and set $b$ to that minimum. Moreover, taking a positive $q(x \mid y)$ on a compact support yields minorization condition for the $q(x \mid y)$.

If the support of target distribution is not compact, we may resort to the approximation of the target distribution on some smaller compact support that contains say 99.9% of the whole mass of target distribution. In practice, many problems of generative modeling are defined on compact support, e.g. the distribution of images lies in finite support since we represent an image by pixels values.

Table 1: Different losses for a density-ratio estimation.

| Proposal | Name | Loss |
|---|---|---|
| Markov | Upper bound (UB) | $\int dx dy \; p(x)q(y\,|\,x)\left[\log\dfrac{d(y,x)}{d(x,y)} + \dfrac{d(y,x)}{d(x,y)}\right]$ |
| | Markov cross-entropy (MCE) | $\int dx dy \; p(x)q(y\,|\,x)[-\log d(x,y) - \log(1 - d(y,x))]$ |
| Independent | Conventional cross-entropy (CCE) | $\int dx dy \; p(x)q(y)[-\log d(x)(1 - d(y))]$ |

### 3.5 Relation to the cross-entropy

It is possible to upper bound the loss (18) by the binary cross-entropy. For a Markov proposal, it is

$$\mathbb{E}_{\substack{x \sim p(x) \\ y \sim q(y\,|\,x)}} \left[\log\frac{d(y,x)}{d(x,y)} + \frac{d(y,x)}{d(x,y)}\right] \leq \mathbb{E}_{\substack{x \sim p(x) \\ y \sim q(y\,|\,x)}} \left[-\log d(x,y) - \log(1 - d(y,x)) + \frac{1}{b}\right]. \quad (22)$$

In the case of an independent proposal, we factorize the discriminator as $d(x,y) = d(x)(1 - d(y))$ and obtain the following inequality (see Appendix F).

$$\mathbb{E}_{\substack{x \sim p(x) \\ y \sim q(y\,|\,x)}} \left[\log\frac{d(y,x)}{d(x,y)} + \frac{d(y,x)}{d(x,y)}\right] \leq -\mathbb{E}_{x\sim p(x)}\log d(x) - \mathbb{E}_{y\sim q(y)}\log(1 - d(y)) + \frac{1}{b} \quad (23)$$

Thus, learning a discriminator via the binary cross-entropy, we also minimize the distance $\|t_\infty - p\|_{TV}$. This fact justifies Algorithm 2.

## 4 Experiments

We present the empirical evaluation of the proposed algorithm and theory for both independent and Markov proposals. For independent proposals, we validate our theoretical result by demonstrating monotonous improvements of the sampling procedure throughout the learning of the discriminator. Further, the implicit MH algorithm with a Markov proposal compares favorably against Algorithm 2 proposed by (Neklyudov et al., 2018; Turner et al., 2018). In both cases, sampling via the implicit MH algorithm always improves over the straightforward sampling from the proposal. Code reproducing all experiments is available online[3].

To assess our theoretical result in practice, we demonstrate that the minimization of the derived upper bounds (17), (22), (23) results in the minimization of the distance between the target distribution and the distribution of the chain. Since one can evaluate the total variation distance only when explicit densities are given, we show its monotonous fall only for synthetic examples (Appendix G). Also, we provide an analysis of the algorithm with the growth of dimensionality (Appendix G).

For complex empirical distributions, we consider the problem of sampling from the space of images (CIFAR-10, CelebA, ImageNet datasets) and resort to the conventional metrics for the performance evaluation: the Inception Score (IS) (Salimans et al., 2016) and Frechet Inception Distance (FID) (Heusel et al., 2017). Note that these metrics rely heavily on the implementation of Inception network (Barratt & Sharma, 2018); therefore, for all experiments, we use PyTorch version of the Inception V3 network (Paszke et al., 2017).

### 4.1 Independent proposals

Since we propose to use the implicit MH algorithm for any generative model learned from the empirical distribution, we consider five models that are learned with completely different objectives: Deep Convolutional GAN (DCGAN) (Radford et al., 2015), Variational Auto-Encoder (VAE) (Kingma & Welling, 2014), Wasserstein GAN with gradient penalty (WPGAN) (Gulrajani et al.,

2017), MMD-GAN (Li et al., 2017), BigGAN (Brock et al., 2018). We take the generative part from each already learned model and treat it as an independent proposal distribution in Algorithm 3. For GANs, we take the generator, for VAE, we take the decoder and the prior. Then we learn the discriminator from scratch for all models (except BigGAN; there we finetune the head of the discriminator) and monitor the performance of the Algorithm 3 with iterations.

Our theoretical result says that the total variation distance between the stationary distribution and the target can be upper bounded by different losses (see Table 1). Note, that we also can learn a discriminator by UB and MCE for independent proposals; however, in practice, we found that CCE performs slightly better. In Figure 8, we demonstrate that the minimization of CCE leads to better IS and FID throughout the learning of a discriminator (see plots for all models in Appendix H). However, for a finite empirical distribution, expressive enough discriminator could overfit to the target dataset. In such a case, the implicit MH algorithm would become infeasible since it would accept only samples that match points of the dataset. This can be averted by monitoring the acceptance rate and early stopping to prevent overfitting.

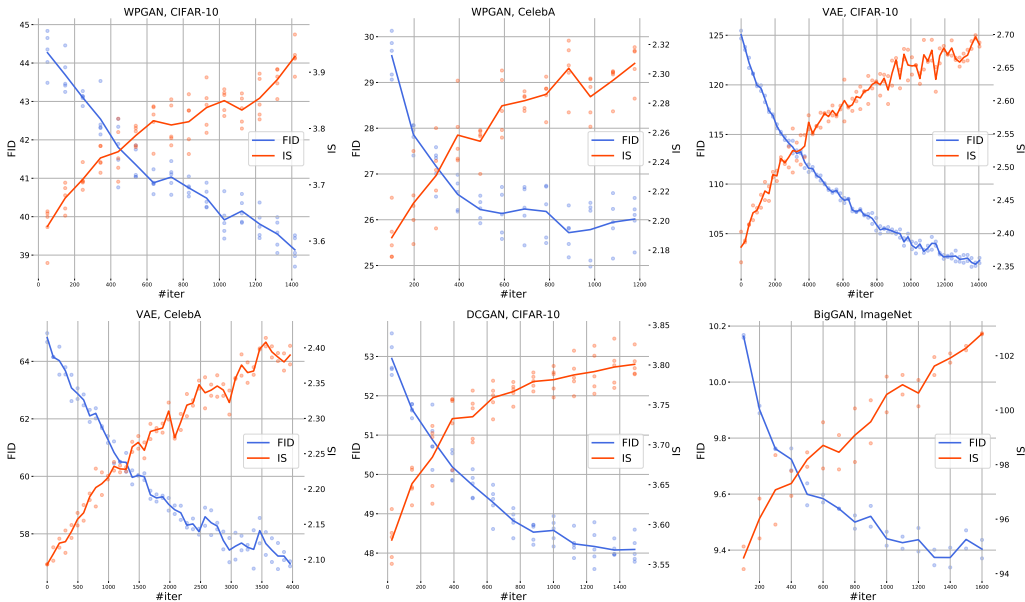

Figure 1: Monotonous improvements in terms of FID and IS for the learning of discriminator by CCE. During iterations, we evaluate metrics several times (scatter) and then average them (solid lines). For a single metric evaluation, we use 10k samples. Higher values of IS and lower values of FID are better. Performance for the original models corresponds to the 0th iteration on the plots.

## 4.2 Markov proposals

To simulate Markov proposals we take the same WPGAN as in the independent case and traverse its latent space by a Markov chain. Taking the latent vector $z_y$ for the previous image $y$, we sample the next vector $z_x$ via HMC and obtain the next image $x = g(z_x)$ by the generator $g(\cdot)$, thus simulating a Markov proposal $q(x \mid y)$. Sampling via HMC from the Gaussian is equivalent to the interpolation between the previous accepted point $z_y$ and the random vector $v$:

$$z_x = \cos(t)z_y + \sin(t)v, \quad v \sim \mathcal{N}(0, I). \tag{24}$$

In our experiments, we take $t = \pi/3$. For loss estimation, we condition samples from the proposal on samples from the dataset $x \sim q(x \mid y), y \sim p(y)$. However, to sample an image $x \sim q(x \mid y)$ we need to know the latent vector $z_y$ for an image $y$ from the dataset. We find such vectors by optimization in the latent space, minimizing the $l_2$ reconstruction error (reconstructions are in Fig. 2).

To filter a Markov proposal, we need to learn a pairwise discriminator, as suggested in Section 3. For this purpose, we take the same architecture of the discriminator as in the independent case and put

the difference of its logits net($\cdot$) into the sigmoid.

$$d(x, y) = \frac{1}{1 + \exp(\text{net}(y) - \text{net}(x))} \tag{25}$$

Then we learn this discriminator by minimization of UB and MCE (see Table 1).

In Figure 3, we demonstrate that our Markov proposal compares favorably not only against the original generator of WPGAN, but also against the chain obtained by the independent sampler (Algorithm 2). To provide the comparison, we evaluate both the performance (IS, FID) and computational efforts (rejection rate), showing that for the same rejection rate, our method results in better metrics.

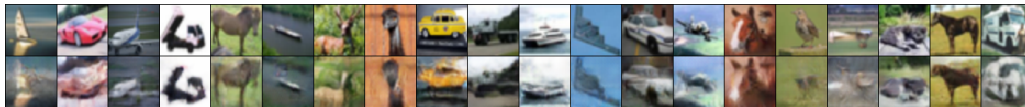

Figure 2: Samples from CIFAR-10 (top line) and their reconstructions (bottom line)

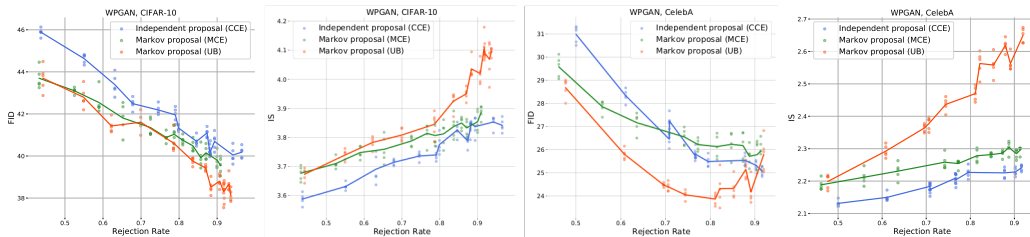

Figure 3: Comparison between different discriminators for the same generator of WPGAN in terms of performance (IS, FID) and computational efforts (rejection rate). Higher values of IS and lower values of FID are better. For a single metric evaluation, we use 10k samples. For every snapshot of a discriminator, we evaluate metrics 5 times (scatter) and then average them (solid lines).

## 5 Conclusion

In this paper, we propose the implicit Metropolis-Hastings algorithm for sampling from the empirical target distribution, assuming that the proposal only able to generate samples (without an access to the density). In the theoretical part of the paper, we upper bound the distance between the target distribution and the stationary distribution of the chain. The contribution of the derived upper bound is two-fold. We justify the heuristic algorithm proposed by (Neklyudov et al., 2018; Turner et al., 2018) and derive the loss functions for the case of Markov proposal. Moreover, the post-processing with the implicit Metropolis-Hastings algorithm can be seen as the theoretical justification of any generative model learned from the empirical target distribution. In the experimental part of the paper, we empirically validate the proposed algorithm on the real-world datasets (CIFAR-10, CelebA, ImageNet) using different generative models as proposals. For all models and datasets filtering via the proposed algorithm alleviates the gap between target and proposal distributions.

## 6 Acknowledgements

This research is in part based on the work supported by Samsung Research, Samsung Electronics. Dmitry Vetrov and Kirill Neklyudov were supported by the Russian Science Foundation grant no. 19-71-30020.

## Footnotes

*National Research University Higher School of Economics

†Skolkovo Institute of Science and Technology

[3]`https://github.com/necludov/implicit-MH`

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
