[Supplementary Material · IMH_NeurIPS2019_appendix.pdf]

# A Basic properties

Let us recall that transition kernel of the implicit Metropolis-Hastings algorithm is defined as

$$t(x \mid y) = q(x \mid y) \min \left\{ 1, \frac{d(x,y)}{d(y,x)} \right\} + \delta(x - y) \int dx' q(x' \mid y) \left( 1 - \min \left\{ 1, \frac{d(x',y)}{d(y,x')} \right\} \right). \quad (26)$$

In this section we show that such kernel has some basic properties that, along with recurrence, imply convergence to some stationary distribution. To show these properties we assume that the proposal distribution $q(x \mid y)$ and the function $d(x, y)$ are *continuous* and *positive* on $\mathbb{R}^D \times \mathbb{R}^D$.

Firstly, we validate that such transition kernel defines a correct conditional distribution.

$$t(x \mid y) \geq q(x \mid y) \min \left\{ 1, \frac{d(x,y)}{d(y,x)} \right\} > 0 \ \forall x, y \implies t_1(x) = \int t(x \mid y) t_0(y) dy > 0 \ \forall x \quad (27)$$

Normalization constant of $t_1$ can be obtained by straightforward evaluation of the integral:

$$t_1(x) = \int dy q(x \mid y) t_0(y) \min \left\{ 1, \frac{d(x,y)}{d(y,x)} \right\} + \quad (28)$$

$$+ \int dy \delta(x - y) t_0(y) \int dx' q(x' \mid y) \left( 1 - \min \left\{ 1, \frac{d(x',y)}{d(y,x')} \right\} \right) \quad (29)$$

$$t_1(x) = \int dy q(x \mid y) t_0(y) \min \left\{ 1, \frac{d(x,y)}{d(y,x)} \right\} + t_0(x) - \quad (30)$$

$$- \int dx' q(x' \mid x) t_0(x) \min \left\{ 1, \frac{d(x',x)}{d(x,x')} \right\} \quad (31)$$

$$\int t_1(x) dx = \int dx dy q(x \mid y) t_0(y) \min \left\{ 1, \frac{d(x,y)}{d(y,x)} \right\} + \int t_0(x) dx - \quad (32)$$

$$- \int dx dx' q(x' \mid x) t_0(x) \min \left\{ 1, \frac{d(x',x)}{d(x,x')} \right\} \quad (33)$$

$$\int t_1(x) dx = \int t_0(x) dx = 1 \quad (34)$$

## A.1 Irreducibility

Irreducibility of the chain can be straightforwardly proven by adaptation of the proof from (Roberts et al., 2004).

Consider some set $A$ such that $p(A) > 0$. Then there exist $R > 0$ such that $p(A_R) > 0$ where $A_R = A \cap B_R(0)$ and $B_R(0)$ is a ball with radius $R$ centered at zero. For continuous and positive $d(x, y)$ and $q(x \mid y)$ on $\mathbb{R}^D \times \mathbb{R}^D$ there exist $\varepsilon > 0$ such that

$$\inf_{x,y \in A_R} q(x \mid y) \min \left\{ 1, \frac{d(x,y)}{d(y,x)} \right\} \geq \inf_{x,y \in B_R} q(x \mid y) \min \left\{ 1, \frac{d(x,y)}{d(y,x)} \right\} \geq \varepsilon. \quad (35)$$

Hence

$$t(A \mid y) \geq t(A_R \mid y) \geq \int_{A_R} q(x \mid y) \min \left\{ 1, \frac{d(x,y)}{d(y,x)} \right\} dx \geq \varepsilon |A_R| > 0. \quad (36)$$

Thus the chain defined by $t(x \mid y)$ is irreducible.

## A.2 Aperiodicity

Aperiodicity of the chain can be straightforwardly proven by adaptation of the proof from (Roberts et al., 2004).

Assume there exist two disjoint sets $A_1$ and $A_2$, such that for any starting point $y \in A_1$ the transition $t(x \mid y)$ ends in $A_2$, i.e. $t(A_2 \mid y) = 1$. However, by positivity of $d(x, y)$ and $q(x \mid y)$ we have

$$t(A_1 \mid y) = \int_{A_1} q(x \mid y) \min \left\{ 1, \frac{d(x,y)}{d(y,x)} \right\} dx > 0 \implies t(A_2 \mid y) < 1. \quad (37)$$

# B  Proof of Proposition 1

We consider some ergodic chain with kernel $t(x\,|\,y)$ and assume that $t(x\,|\,y)$ satisfy minorization condition, i.e. for some distribution $\nu$ and some $\varepsilon > 0$ the following inequality holds:

$$t(x\,|\,y) \geq \varepsilon\nu(x), \quad \forall x, y. \tag{38}$$

We denote a distribution after $n$ steps of $t(x\,|\,y)$ as $t_n(x\,|\,y)$. Such distribution is defined by the recurrent formula:

$$t_{n+1}(x) = \int t(x\,|\,y)t_{n-1}(y)dy. \tag{39}$$

Denoting the difference between two consequent distributions as $\Delta_n$, we study how the operator $t(x\,|\,y)$ changes the $l_1$-norm of $\Delta_n$.

$$t_{n+1}(y) = t_n(y) + \Delta_n(y) \implies \int t(x\,|\,y)t_{n+1}(y)dy = \int t(x\,|\,y)t_n(y)dy + \int t(x\,|\,y)\Delta_n(y)dy \tag{40}$$

Therefore

$$\|t_{n+1} - t_n\|_{TV} = \frac{1}{2}\int |\Delta_n(y)|dy, \quad \text{and} \quad \|t_{n+2} - t_{n+1}\|_{TV} = \frac{1}{2}\int \left|\int t(x\,|\,y)\Delta_n(y)dy\right|dx. \tag{41}$$

Note that $\Delta_n$ integrates in zero

$$\int \Delta_n(y)dy = \int t_{n+1}(y)dy - \int t_n(y)dy = 0. \tag{42}$$

Using that fact we can rewrite the following integral

$$\int t(x\,|\,y)\Delta_n(y)dy = \int (t(x\,|\,y) - \varepsilon\nu(x))\Delta_n(y)dy \tag{43}$$

$$\frac{1}{2}\int \left|\int t(x\,|\,y)\Delta_n(y)dy\right|dx \leq \frac{1}{2}\int (t(x\,|\,y) - \varepsilon\nu(x))|\Delta_n(y)|dydx = (1-\varepsilon)\frac{1}{2}\int |\Delta_n(y)|dy \tag{44}$$

Using the last inequality and equalities from (41), we obtain

$$\|t_{n+2} - t_{n+1}\|_{TV} \leq (1-\varepsilon)\|t_{n+1} - t_n\|_{TV}. \tag{45}$$

# C  Proof of Proposition 2

For the kernel of implicit Metropolis-Hastings algorithm:

$$t(x\,|\,y) = q(x\,|\,y)\min\left\{1, \frac{d(x,y)}{d(y,x)}\right\} + \delta(x-y)\int dx' q(x'\,|\,y)\left(1 - \min\left\{1, \frac{d(x',y)}{d(y,x')}\right\}\right), \tag{46}$$

we want to derive upper bound on the length of the first step in terms of TV-distance

$$\|t_1 - p\|_{TV} = \frac{1}{2}\int dx\left|\int dy t(x\,|\,y)p(y) - p(x)\right|. \tag{47}$$

Firstly, we take the integral inside of TV-distance:

$$\int dy t(x\,|\,y)p(y) = \int dy q(x\,|\,y)p(y)\min\left\{1, \frac{d(x,y)}{d(y,x)}\right\} + \int dy\delta(x-y)p(y) - \tag{48}$$

$$- \int dx'dy\delta(x-y)q(x'\,|\,y)p(y)\min\left\{1, \frac{d(x',y)}{d(y,x')}\right\} = \tag{49}$$

$$= \int dy q(x\,|\,y)p(y)\min\left\{1, \frac{d(x,y)}{d(y,x)}\right\} + p(x) - \int dx'q(x'\,|\,x)p(x)\min\left\{1, \frac{d(x',x)}{d(x,x')}\right\} = \tag{50}$$

$$= \int dy q(x\,|\,y)p(y)\min\left\{1, \frac{d(x,y)}{d(y,x)}\right\} + p(x) - \int dy q(y\,|\,x)p(x)\min\left\{1, \frac{d(y,x)}{d(x,y)}\right\} \tag{51}$$

Substituting this formula into (47) we obtain

$$\|t_1 - p\|_{TV} = \frac{1}{2} \int dx \left| \int dy q(x \mid y) p(y) \min \left\{ 1, \frac{d(x,y)}{d(y,x)} \right\} - \int dy q(y \mid x) p(x) \min \left\{ 1, \frac{d(y,x)}{d(x,y)} \right\} \right| \le \tag{52}$$

$$\le \frac{1}{2} \int dx dy \left| q(x \mid y) p(y) \min \left\{ 1, \frac{d(x,y)}{d(y,x)} \right\} - q(y \mid x) p(x) \min \left\{ 1, \frac{d(y,x)}{d(x,y)} \right\} \right| = \tag{53}$$

Note that changing variables in integral does not change value of function, hence we can integrate over the half of the space and then multiply the integral by 2:

$$= \int_A dx dy \left| q(x \mid y) p(y) \min \left\{ 1, \frac{d(x,y)}{d(y,x)} \right\} - q(y \mid x) p(x) \min \left\{ 1, \frac{d(y,x)}{d(x,y)} \right\} \right| = \tag{54}$$

$$A = \left\{ x, y : \frac{d(x,y)}{d(y,x)} \ge 1 \right\} \tag{55}$$

$$= \int_A dx dy \left| q(x \mid y) p(y) - q(y \mid x) p(x) \min \left\{ 1, \frac{d(y,x)}{d(x,y)} \right\} \right| \tag{56}$$

Thus, we obtain

$$\|t_1 - p\|_{TV} \le 2 \left\| q(x \mid y) p(y) - q(y \mid x) p(x) \min \left\{ 1, \frac{d(y,x)}{d(x,y)} \right\} \right\|_{TV} \tag{57}$$

## D   Proof of Proposition 3

To prove Proposition 3 we extend the proof from (Pollard, 2000). Consider a distribution $\alpha(x)$ and some positive function $f(x) > 0 \;\; \forall x$. Normalization constants for $\alpha$ and $f$ are

$$\int \alpha(x) dx = 1, \quad \text{and} \quad \int f(x) dx = C. \tag{58}$$

The proof is constructed around the following inequality

$$(1 + r) \log(1 + r) - r \ge \frac{1}{2} \frac{r^2}{1 + r/3}, \quad r \ge -1. \tag{59}$$

For $r$ we consider the ratio $r(x) = \alpha(x)/f(x) - 1$, and introduce a random variable $F$ with the density $f(x)/C$. Then

$$\mathbb{E}_F r(x) = \int \frac{f(x)}{C} \left( \frac{\alpha(x)}{f(x)} - 1 \right) dx = \frac{1}{C} - 1 \tag{60}$$

$$\mathbb{E}_F (1 + r(x)) \log(1 + r(x)) = \frac{1}{C} \int \alpha(x) \log \frac{\alpha(x)}{f(x)} \triangleq \frac{1}{C} \widehat{\mathrm{KL}}(\alpha \| f) \tag{61}$$

$$\mathbb{E}_F \left( 1 + \frac{r(x)}{3} \right) = \frac{2}{3} + \frac{1}{3C} > 0 \tag{62}$$

$$\mathbb{E}_F |r(x)| = \frac{1}{C} \int \left| \alpha(x) - f(x) \right| dx = \frac{2}{C} \|\alpha - f\|_{TV} \tag{63}$$

Substituting all the equations into (59) we obtain

$$\mathbb{E}_F \left[ (1 + r(x)) \log(1 + r(x)) - r(x) \right] \ge \frac{1}{2} \mathbb{E}_F \left[ \frac{r(x)^2}{1 + r(x)/3} \right] \tag{64}$$

$$\mathbb{E}_F \left( 1 + \frac{r(x)}{3} \right) \mathbb{E}_F \left[ (1 + r(x)) \log(1 + r(x)) - r(x) \right] \ge \frac{1}{2} \mathbb{E}_F \left[ \frac{r(x)^2}{1 + r(x)/3} \right] \mathbb{E}_F \left( 1 + \frac{r(x)}{3} \right) \tag{65}$$

$$\mathbb{E}_F \left( 1 + \frac{r(x)}{3} \right) \mathbb{E}_F \left[ (1 + r(x)) \log(1 + r(x)) - r(x) \right] \ge \frac{1}{2} \left[ \mathbb{E}_F |r(x)| \right]^2 \tag{66}$$

$$\frac{2C + 1}{3C} \left( \frac{1}{C} \widehat{\mathrm{KL}}(\alpha \| f) - \frac{1}{C} + 1 \right) \ge \frac{2}{C^2} \|\alpha - f\|_{TV}^2 \tag{67}$$

Hence, we obtain

$$\|\alpha - f\|_{TV}^2 \leq \frac{2C+1}{6}\left(\widehat{\mathrm{KL}}(\alpha\|f) + C - 1\right) \tag{68}$$

Note, that if $f$ is a distribution, then $C = 1$ and we obtain Pinsker's inequality:

$$\|\alpha - f\|_{TV}^2 \leq \frac{1}{2}\widehat{\mathrm{KL}}(\alpha\|f). \tag{69}$$

# E  DRE

We derive the formula for the optimal discriminator by taking derivative of the following objective w.r.t. the value of $d(x, y)$ in a single point $(x, y)$

$$\min_d \mathbb{E}_{\substack{x \sim p(x) \\ y \sim q(y\,|\,x)}} \left[\log \frac{d(y, x)}{d(x, y)} + \frac{d(y, x)}{d(x, y)}\right]. \tag{70}$$

Speaking informally, we treat the expectation as a sum over all the possible points. Taking a derivative w.r.t. a single point allows us to consider only two elements of the sum.

$$\nabla_{d(x,y)}\left(p(x)q(y\,|\,x)\left[\log \frac{d(y,x)}{d(x,y)} + \frac{d(y,x)}{d(x,y)}\right] + p(y)q(x\,|\,y)\left[\log \frac{d(x,y)}{d(y,x)} + \frac{d(x,y)}{d(y,x)}\right]\right) = 0 \tag{71}$$

$$p(x)q(y\,|\,x)\left[-\frac{1}{d(x,y)} - \frac{d(y,x)}{d(x,y)^2}\right] + p(y)q(x\,|\,y)\left[\frac{1}{d(x,y)} + \frac{1}{d(y,x)}\right] = 0 \tag{72}$$

$$\frac{p(x)q(y\,|\,x)}{p(y)q(x\,|\,y)}\left[-1 - \frac{d(y,x)}{d(x,y)}\right] + \left[1 + \frac{d(x,y)}{d(y,x)}\right] = 0 \tag{73}$$

$$\frac{p(x)q(y\,|\,x)}{p(y)q(x\,|\,y)}\frac{d(y,x) + d(x,y)}{d(x,y)} = \frac{d(x,y) + d(y,x)}{d(y,x)} \tag{74}$$

$$\frac{p(x)q(y\,|\,x)}{p(y)q(x\,|\,y)} = \frac{d(x,y)}{d(y,x)} \tag{75}$$

Note that we do not derive an explicit form of $d(x, y)$, actually $d(x, y)$ could be any function which ratio equals to the density-ratio.

The same result can be obtained by taking a derivative in function space, but for simplicity, we provide here an informal proof by taking the pointwise derivative.

# F  Relation to the cross-entropy

In this Section we make a connection between loss (18) and the conventional loss for the density-ratio estimation — cross-entropy.

## F.1  Markov proposal

For Markov proposal, the loss from (18) can be straightforwardly upper bounded by the cross-entropy:

$$\mathbb{E}_{\substack{x \sim p(x) \\ y \sim q(y\,|\,x)}}\left[\log \frac{d(y,x)}{d(x,y)} + \frac{d(y,x)}{d(x,y)}\right] \leq \mathbb{E}_{\substack{x \sim p(x) \\ y \sim q(y\,|\,x)}}\left[-\log d(x,y) - \log(1 - d(y,x)) + \frac{1}{b}\right]. \tag{76}$$

That yields the optimal discriminator

$$d(x, y) = \frac{p(x)q(y\,|\,x)}{p(x)q(y\,|\,x) + p(y)q(x\,|\,y)}, \tag{77}$$

using which we can achieve $\|t_\infty - p\|_{TV} = 0$.

### F.2 Independent proposal

In Section 2 we describe Algorithm 2 proposed in (Neklyudov et al., 2018; Turner et al., 2018). The idea of the algorithm is to use learned generator of any GAN model as *independent* proposal $q(x)$ in the Metropolis-Hastings algorithm. Authors propose to learn a discriminator $d(x)$ by minimization of the cross-entropy:

$$\min_d \left[ -\mathbb{E}_{x \sim p(x)} \log d(x) - \mathbb{E}_{x \sim q(x)} \log(1 - d(x)) \right], \tag{78}$$

and then to estimate the density-ratio as

$$\frac{p(x)q(y)}{p(y)q(x)} \approx \frac{d(x)(1 - d(y))}{(1 - d(x))d(y)}. \tag{79}$$

In this section, we show that there exists such an upper bound on $\|t_\infty - p\|_{TV}$ that its optimization is equivalent to the optimization of cross-entropy (78). To derive such upper bound we upper bound the discriminator objective (18), considering an independent proposal $q(x)$ and factorized discriminator $d(x, y) = d(x)(1 - d(y))$.

$$\mathbb{E}_{\substack{x \sim p(x) \\ y \sim q(y)}} \left[ \log \frac{d(y)(1 - d(x))}{d(x)(1 - d(y))} + \frac{d(y)(1 - d(x))}{d(x)(1 - d(y))} \right] \leq \mathbb{E}_{\substack{x \sim p(x) \\ y \sim q(y)}} \left[ \log \frac{d(y)(1 - d(x))}{d(x)(1 - d(y))} + \frac{1}{b} \right] \tag{80}$$

Splitting the logarithm into sum results in

$$\left[ -\mathbb{E}_{x \sim p(x)} \log d(x) - \mathbb{E}_{y \sim q(y)} \log(1 - d(y)) + \mathbb{E}_{x \sim p(x)} \log(1 - d(x)) + \mathbb{E}_{y \sim q(y)} \log d(y) \right] \leq$$
$$\leq -\mathbb{E}_{x \sim p(x)} \log d(x) - \mathbb{E}_{y \sim q(y)} \log(1 - d(y)), \tag{81}$$

where the last upper bound is the cross-entropy (78). The obtained upper bound on the discriminator objective (18) can be substituted in (17) that results in

$$\|t_\infty - p\|_{TV}^2 \leq \mathcal{L}(d) \leq \left( \frac{4 + 2b}{3\varepsilon^2 b^3} \right) \cdot$$
$$\cdot \left( -\mathbb{E}_{x \sim p(x)} \log d(x) - \mathbb{E}_{y \sim q(y)} \log(1 - d(y)) + \frac{1}{b} - 1 + \mathrm{KL}(q(y)p(x) \| q(x)p(y)) \right). \tag{82}$$

Hence, minimization of the cross-entropy leads to the minimization of the TV-distance between stationary distribution of the chain $t_\infty(x)$ and target distribution $p(x)$. Note that during optimization of such upper-bound we also could achieve $\|t_\infty - p\|_{TV} = 0$ for any target $p(x)$ and proposal $q(x)$, since the optimal discriminator $d^*(x)$ allows correct estimation of density ratio:

$$\frac{d^*(x)(1 - d^*(y))}{(1 - d^*(x))d^*(y)} = \frac{p(x)q(y)}{p(y)q(x)}. \tag{83}$$

## G   Synthetic examples

### G.1   1-D mixture of Gaussians

We validate the proposed algorithm and compare different losses on the synthetic target distribution. For the target empirical distribution we take 5000 samples from the mixture of two Gaussians $p(x) = 0.5\mathcal{N}(x \,|\, \mu = -2, \sigma = 0.5) + 0.5\mathcal{N}(x \,|\, \mu = 2, \sigma = 0.7)$. We simulate an implicit Markov proposal by sampling from the random-walk kernel $q(x \,|\, y) = \mathcal{N}(x \,|\, \mu = y, \sigma = 1.0)$, and an implicit independent proposal by sampling from the Gaussian $q(x) = \mathcal{N}(x \,|\, \mu = 0.0, \sigma = 2.0)$. Note, despite that we know densities of the target and proposals, we use only samples from these distributions during training and sampling stages. As a discriminator, we use the neural network with 3 fully-connected layers (100 hidden neurons) and learn it with the Adam optimizer for 1000 iterations.

Since we have access to the density of distributions, we monitor the TV-distance (from (10))

$$2\left\|q(y\,|\,x)p(x) - q(x\,|\,y)p(y)\frac{d(x,y)}{d(y,x)}\right\|_{TV} \tag{84}$$

throughout the learning of the discriminator. This distance can be treated as the averaged $l_1$-error of the density-ratio estimation:

$$2\left\|q(y\,|\,x)p(x) - q(x\,|\,y)p(y)\frac{d(x,y)}{d(y,x)}\right\|_{TV} = \int dxdy\; q(x\,|\,y)p(y)\left|\frac{q(y\,|\,x)p(x)}{q(x\,|\,y)p(y)} - \frac{d(x,y)}{d(y,x)}\right|. \tag{85}$$

We compare losses from Table 1 in Figure 4. In agreement with our theoretical result, the TV-distance steadily goes down during the learning of the discriminator. For Markov proposal (left plot in Fig. 4), the optimization of upper bound (UB) behaves similarly to the optimization of the cross-entropy (MCE). However, for the independent proposal (right plot in Fig. 4), the best metric for optimization is the conventional cross-entropy (CCE). In Figure 6, we demonstrate filtering of the independent proposal with the discriminator learned by the optimization of the cross-entropy (CCE).

Note that learning a discriminator for the random-walk proposal allows for estimation of target unnormalized density:

$$\frac{d(x,y)}{d(y,x)} \approx \frac{p(x)q(y\,|\,x)}{p(y)q(x\,|\,y)} = \frac{p(x)}{p(y)}, \tag{86}$$

since $q(x\,|\,y) = q(y\,|\,x)$.

Figure 4: Comparison of different losses for a discriminator in terms of the TV-distance (84). On the left plot we learn the discriminator for the Markov proposal, on the right plot we learn the discriminator for the independent proposal. For losses see Table 1.

Figure 5: Samples from the independent proposal distribution are on the left. Samples obtained after filtering with the implicit Metropolis-Hastings (IMH) algorithm are on the right.

## G.2 Scaling with dimensions

To assess the behavior of the algorithm on high-dimensional tasks, we consider a tractable toy distribution and gradually increase its number of dimensions. As in (Roberts et al., 2001), for the target, we take factorized distribution $p(x) = p(x_1)\prod_{i=2}^{d} p(x_i)$, where $p(x_1) = 0.5\mathcal{N}(x\,|\,\mu = -1.5, \sigma = 0.5) + 0.5\mathcal{N}(x\,|\,\mu = 1.5, \sigma = 0.7)$ is the mixture of two Gaussians and the rest $d-1$ components are standard normal $p(x_i) = \mathcal{N}(0,1)$. For the Markov proposal, we take the homogeneous random-walk kernel $q(x|y) = \mathcal{N}(x|y, \sigma I)$ and scale $\sigma$ with dimensions as proposed in (Roberts et al., 2001) to keep the acceptance rate about 20%. For the independent proposal, we take homogeneous Gaussian $q(x) = \mathcal{N}(0, \sigma I), \sigma = 1.2$.

To evaluate the quality of a chain we draw 20000 samples from the chain and compute the symmetric KL-divergence along the first dimension (as the most difficult), averaging across 100 independent runs. Since for the symmetric KL we need both densities, we approximate the density of the samples by a histogram. In Fig. 6, we demonstrate the quality of sampling with the growth of dimensionality. For the reference performance we take the exact Metropolis-Hastings algorithm (MH on the plots). The implicit MH algorithm demonstrate a comparable performance with the exact MH algorithm for all loss functions of the discriminator. This fact indicates that the growth of the symmetric KL divergence is a property of the MH algorithm itself (not a poor test approximation). In other words, the error in the test is negligible compared to the error induced by the growth of dimensionality.

Figure 6: The mean error of the chains measured as the symmetric KL divergence along the first dimension of the target distribution. We average across 100 independent runs and plot the variance for each dimension. The error of the implicit MH algorithm (different losses: MCE, UB, CCE) grows comparably with the error of the exact MH algorithm. For the density of the target distribution see details in the text.

# H    Monotonous improvements throughout the learning of the discriminator

Figure 7: Monotonous improvements in terms of FID and IS for the learning of discriminator by CCE. During iterations, we evaluate metrics several times (scatter) and then average them (solid lines). For a single metric evaluation, we use 10k samples. Higher values of IS and lower values of FID are better. Performance for the original models corresponds to the 0th iteration on the plots.

Figure 8: Monotonous improvements in terms of FID and IS for the learning of discriminator by CCE. During iterations, we evaluate metrics several times (scatter) and then average them (solid lines). For a single metric evaluation, we use 10k samples. Higher values of IS and lower values of FID are better. Performance for the original models corresponds to the 0th iteration on the plots.

# I   Losses for DRE

Table 2: Different losses for the density-ratio estimation.

| PROPOSAL | NAME | LOSS | DRE |
|---|---|---|---|
| MARKOV | UPPER BOUND (UB) | $\int dxdy\, p(x)q(y\mid x)\left[\log\frac{d(y,x)}{d(x,y)}+\frac{d(y,x)}{d(x,y)}\right]$ | $\frac{p(x)q(y\mid x)}{p(y)q(x\mid y)}=\frac{d(x,y)}{d(y,x)}$ |
| | MARKOV CROSS-ENTROPY (MCE) | $\int dxdy\, p(x)q(y\mid x)[-\log d(x,y)-\log(1-d(y,x))]$ | $\frac{p(x)q(y\mid x)}{p(y)q(x\mid y)}=\frac{d(x,y)}{d(y,x)}$ |
| | LINEAR TERM (LT) | $\int dxdy\, p(x)q(y\mid x)\left[\frac{d(y,x)}{d(x,y)}\right]$ | $\frac{p(x)q(y\mid x)}{p(y)q(x\mid y)}=\left(\frac{d(x,y)}{d(y,x)}\right)^2$ |
| INDEPENDENT | UPPER BOUND (UB) | $\int dxdy\, p(x)q(y)\left[\log\frac{d(y)(1-d(x))}{d(x)(1-d(y))}+\frac{d(y)(1-d(x))}{d(x)(1-d(y))}\right]$ | $\frac{p(x)q(y)}{p(y)q(x)}=\frac{d(x)(1-d(y))}{d(y)(1-d(x))}$ |
| | MARKOV CROSS-ENTROPY (MCE) | $\int dxdy\, p(x)q(y)[-\log d(x)(1-d(y))-\log(1-d(y)(1-d(x)))]$ | $\frac{p(x)q(y)}{p(y)q(x)}=\frac{d(x)(1-d(y)(1-d(x)))}{(1-d(x))(1-d(x)(1-d(y)))}\cdot\frac{(2(1-d(x)(1-d(y)))-d(y))}{((1-d(y))(1-d(x)))+d(y)d(x)}$ |
| | LINEAR TERM (LT) | $\int dxdy\, p(x)q(y)\left[\frac{d(y)(1-d(x))}{d(x)(1-d(y))}\right]$ | $\frac{p(x)q(y)}{p(y)q(x)}=\left(\frac{d(x)(1-d(y))}{d(y)(1-d(x))}\right)^2$ |
| | CONVENTIONAL CROSS-ENTROPY (CCE) | $\int dxdy\, p(x)q(y)[-\log d(x)(1-d(y))]$ | $\frac{p(x)q(y)}{p(y)q(x)}=\frac{d(x)(1-d(y))}{d(y)(1-d(x))}$ |