[Reviews · NeurIPS 2019]

Reviewer 1



- L109 "These properties imply convergence of the Markov chain defined by t to some stationary distribution": in general, irreducibility and aperiodicity does not imply the existence of a stationary distribution (eg. a standard Random walk). - the TV metric does not seem like a good metric for measuring convergence in very high-dimensional spaces such as probability distributions on the set of natural images (where GANs are usually leveraged). - assumption in Section 3.1 are terribly restrictive (especially in high-dim settings) and are not likely to be satisfied in any realistic scenarios where GANs are usually used. - the paper seems to use the languages of convergence of Markov Chain to motivate an additional "post-processing" step on top of a standard GAN / generative process. This is definitely interesting, although the derivation through minoration conditions / cvg of MC is not that convincing. - Although the derivation is not convincing, the numerical seem to indicate that the resulting methodology is worthwhile.

Reviewer 2



1. The paper proposes the implicit MH algorithm, which can be used to improve performance of implicit models like GANs. The proposed method can be applied to different models in practice. Compared to some previous work, the proposed method doesn’t need the optimality of the discriminator. 2. The paper provides some analysis on how to minimize the total variation distance between stationary distribution of the chain and the target distribution, how the different objective functions are derived. It also shows some relationships between the proposed method and previous works. 3. The paper considers both Markov proposal and independent proposal, shows how the proposed method can be applied in these scenarios with different objective functions. 4, The motivation is not very clear. The setting seems to suggest applications for sample generation based on a training dataset, similar to the GAN setting. Although experiments show some interesting results, the metric results (FID, IS) are not comparable to some of the recent models (MMD-GAN, SMMD-GAN, Sobolev-GAN, etc.). I think comparisons with some state-of-the-art models are needed; Baselines for comparisons to all the cases are also needed.

Reviewer 3



This paper introduces an algorithm for performing Metropolis-Hastings sampling with implicit distributions. This paper is very well written and the more difficult theory parts of the paper are explained very clearly. I particularly found the bound computed without the use of an optimal discriminator unique and interesting. I consider it a useful contribution to the field of deep generative modeling. In section 4.1, are the models pretrained with their original objective? VAEs (as proposed) are also not implicit models, and you can actually generate samples “correct” samples from them with annealed importance sampling. Would these samples have a 100% accept rate? In general, the experiment setup could be explained in more detail. For example, do you train an encoder for the VAE?

[Author Response · NeurIPS 2019]

We thank all of the reviewers for their valuable feedback and detailed comments. According to the reviewers' sugges-
tions, we want to clarify the main idea of the paper. That is "improvement and justification of any implicit sampler".
We know that in practice, even state-of-the-art generative models yield "unrealistic" samples, hence, are biased. From
the MCMC perspective, we could treat these (already learned) models as proposals for the approximate MH-algorithm
(Algorithm 3). Based on our theoretical analysis, we derive different losses for the discriminator (Table 1 in the paper).
**R1: "irreducibility and aperiodicity does not imply the existence of a stationary distribution . . . "**
Thanks for pointing out this mistake! Indeed, we also need the minorization condition on the transition kernel.
**R1: "the TV metric does not seem like a good metric"**
We assume that the proposed algorithm could be applied not only for images. Hence, we provide analysis in the most
general setting and use the TV-distance as a standard metric in MCMC analysis (Roberts, 2004, general MCMC).
**R1: "assumption in Section 3.1 are terribly restrictive"**
Although the minorization condition for the proposal distribution is indeed restricting, it automatically holds for an
independent proposal (as we note on lines 134-135), which is the most common scenario for GANs. Moreover, if the
support of the target distribution is a compact and the density of the proposal is continuous and positive on this compact,
then we can lower bound the proposal density by a positive constant, hence, satisfy the minorization condition (as we
note on lines 195-197). We can define a distribution of images on the support $[0, 1]^d$ by adding a low-variance Gaussian
noise (truncated to $[0, 1]^d$) to the observations, thus defining positive target and proposal distributions on the compact.
**R1: "empirically test the resulting methodology on tractable high-dimensional toy problems"**
As you suggested, we provide additional experiments for high-dimensional tractable toy problem. As in
(Roberts, 2001, optimal scaling), for the target, we take factorized distribution $p(x) = p(x_1) \prod_{i=2}^{d} p(x_i)$, where
$p(x_1)$ is the mixture of two Gaussians and the rest $d - 1$ components are standard normal $p(x_i) = \mathcal{N}(0, 1)$.
For the Markov proposal, we take homogeneous random-walk $q(x|y) = \mathcal{N}(x|y, \sigma I)$ and scale $\sigma$ with dimen-
sions as proposed in (Roberts, 2001, optimal scaling) to keep the acceptance rate about 20%. For the indepen-
dent proposal, we take homogeneous Gaussian $q(x) = \mathcal{N}(0, \sigma I), \sigma = 1.2$. Empirical results are in Fig. 1.

Figure 1: We evaluate the symmetric KL along the first di-
mension (as the most difficult) for a chain of length 20000,
averaging across 100 independent runs. We compare the
performance of our algorithm for different losses with the
exact MH algorithm (MH on the plots).

Figure 2: Monotonous improvements in terms of FID
and IS for BigGAN and MMD-GAN (both for Markov
and independent proposals). We learn a discriminator for
the Markov proposal by optimization of the Upper Bound.
Performance for the original model (baseline) corresponds
to 0-th iteration of a discriminator.

**R2: "Baselines for comparisons to all the cases are also needed"**
The baselines for our algorithm are the initial implicit models that we improve. In Fig. 1 of the paper, the performance
of the initial model corresponds to the very beginnings of the plots (0-th iteration of learning a discriminator for the
MH-test). Further, we extend the algorithm to the case of Markov proposal. In Fig. 2 of the paper, we demonstrate
that MH with a Markov proposal not only improves the initial models/baselines (the first points on the plots) but also
improves over the independent MH with the same generator network.
**R2: "comparisons with state-of-the-art models are needed"**
Note that we use PyTorch implementation of the InceptionV3 network; hence, the metrics could be different from the
TensorFlow implementation. For instance, the IS for our WGAN varies as 3.6 (PyTorch), 4.7 (TF). As you suggested,
we provide additional experiments for the state-of-the-art models (see Fig. 2).
**R3: "In section 4.1, are the models pretrained with their original objective?"**
Yes, all the models are already trained with their original objective, and we filter them by running Algorithm 3.
**R3: "generate "correct" samples from VAE with AIS. Would these samples have a 100% accept rate?"**
To perform the AIS, one needs the densities of target and proposal. In the case of VAE, we can estimate the density of
the proposal as $q(x) = \mathbb{E}_{p(z)} \text{decoder}(x|z)$, but we still need the density of the target. If we use the same discriminator
for its estimation in AIS, then yes, we will obtain approximately 100% acceptance rate.
**R3: "the experiment setup could be explained in more detail. Do you train an encoder for the VAE?"**
Yes, we train the VAE as in the original paper, then we use only the decoder as a generator by sampling the latent
variables from the prior. We will clarify the experimental setup in the final version.

[Meta-Review · NeurIPS 2019]

This paper introduces an algorithm for performing Metropolis-Hastings sampling with implicit distributions. All three reviewers found the work to be sufficiently interesting and novel, and felt that the empirical analysis suggests the method is promising, viewing the paper overall as borderline but positive. The author response addressed some of the review concerns, as well. NeurIPS is quite competitive but hopefully there will be space to accept this one.